# Consumption of Ultra-Processed Foods and Metabolic Parameters in Type 2 Diabetes Mellitus: A Cross-Sectional Study

**DOI:** 10.3390/ijerph22081275

**Published:** 2025-08-14

**Authors:** Julia Bauer, Fernanda Oliveira Ayala, Aline Marcadenti, Rachel Helena Vieira Machado, Ângela Cristina Bersch-Ferreira, Maria Fernanda Souza Moreira, Mileni Vanti Beretta, Ana Maria Pandolfo Feoli, Fernanda Michielin Busnello

**Affiliations:** 1Graduate Program in Nutrition Sciences, Federal University of Health Sciences of Porto Alegre (UFCSPA), Porto Alegre 90050-170, RS, Brazil; bauerjuulia@gmail.com (J.B.); fernandaayala.nutri@gmail.com (F.O.A.); mfermoreira@gmail.com (M.F.S.M.); mileninutri@gmail.com (M.V.B.); 2Hcor Research Institute (IP-Hcor), São Paulo 04004-030, SP, Brazil; amarcaden@hcor.com.br (A.M.); rhelena@ext.hcor.com.br (R.H.V.M.); angela.bersch@ext.bp.org.br (Â.C.B.-F.); 3School of Health and Life Sciences, Pontifical Catholic University of Rio Grande do Sul (PUCRS), Porto Alegre 90619-900, RS, Brazil; anafeoli@pucrs.br; 4Nutrition Department, Federal University of Health Sciences of Porto Alegre (UFCSPA), Porto Alegre 90050-170, RS, Brazil

**Keywords:** type 2 diabetes mellitus, ultra-processed foods, NOVA food classification

## Abstract

Background: Understanding how food processing impacts type 2 diabetes (T2DM) control is essential for disease management. This study aimed to assess the association between ultra-processed food (UPF) consumption, as defined by the NOVA classification, and metabolic parameters in T2DM patients. Methods: This was a cross-sectional analysis using baseline data from the NUGLIC study, a multicenter randomized clinical trial. Multiple linear and Poisson regressions were used to evaluate associations between quintiles of processed and ultra-processed food consumption and glycated hemoglobin (HbA1c) as the primary outcome. Secondary outcomes included fasting glucose, lipid profile, body mass index (BMI), and waist circumference. Results: This study included 326 participants. UPF consumption accounted for approximately 16.4% of total daily energy intake. No significant linear associations were observed between higher consumption of industrialized foods and anthropometric or glycemic markers. However, intermediate and high consumption levels were associated with an increased total cholesterol (Q3: β = 26.6; Q4: β = 26.7) and LDL-cholesterol (Q4: β = 19.8; Q5: β = 17.5). Conclusion: In T2DM patients, a higher intake of processed and ultra-processed foods was linked to elevated cholesterol and LDL levels. These findings highlight potential cardiovascular risks but do not support causality due to the study’s cross-sectional design.

## 1. Introduction

The global increase in diabetes cases has become a major public health concern, affecting 537 million adults worldwide in 2021, with 80% of them living in low- and middle-income countries [1]. Type 2 diabetes mellitus (T2DM) makes up more than 90% of these cases, causing both large and small blood vessel complications, decreased life expectancy, and a higher risk of heart disease, kidney problems, and cancer [2]. Despite these challenges, it is believed that lifestyle changes like adopting a healthy diet, being more physically active, and managing weight can help prevent or delay the onset of T2DM [3,4,5]. Similarly, for those already diagnosed with T2DM, lifestyle adjustments are crucial alongside medications and education about health, highlighting the importance of a balanced diet in reaching treatment goals and reducing the risk of complications [6,7,8].

In recent years, the concept of diet quality has evolved to encompass more than just the nutritional content of foods, extending to include the level of industrial processing involved in their production [9]. Industrial formulations composed of refined food substances and additives have given rise to a distinct food category known as ultra-processed foods (UPFs). These products are typically ready-to-eat or semi-prepared items with a high energy density, nutritional imbalance, and a lack of bioactive compounds [10]. UPF, along with three other food categories, fresh and minimally processed foods, culinary ingredients, and processed foods (PFs), are classified according to the extent and purpose of industrial processing in the NOVA Food Classification proposed by Monteiro et al. [11].

The proportion of UPF in the diet can be viewed as a population-based metric of diet quality [10], constituting up to 58% of daily energy intake and accounting for 89.7% of total free sugar consumption for more than half of children and adults in developed countries. Among individuals aged 40 to 69, followed for approximately 5 years, consumption of UPF comprising about 22% of daily calories was associated with a 44% higher risk of developing T2DM compared to those with lower daily consumption [12,13]. Further evidence corroborates this finding and suggests that high UPF consumption is also linked to an increased risk of abdominal obesity in adults and older individuals, as well as a higher risk of hypertension in the same age group [14]. The elevated energy density of these foods has been shown to contribute to excess daily energy intake, directly impacting weight gain [15]. Moreover, the heightened level of processing is associated with higher glycemic levels and reduced satiety [16], both important factors in the dietary management of T2DM.

Despite the well-documented relationship between the consumption of UPF and the risk of T2DM, there is currently a lack of evidence regarding the impact of this food group on the treatment of the disease. This knowledge is also crucial for preventing disease progression and reducing mortality in T2DM. It is already known that, in this patient group, higher consumption of UPF, compared to lower consumption, has been associated with a 64% and 155% higher risk of all-cause and cardiovascular mortality, respectively [17]. Therefore, it is important to investigate the consumption of UPF and understand its relationship with the metabolic control parameters of T2DM. Our hypothesis is that individuals who primarily consume diets consisting of UPF may exhibit poorer metabolic parameters. Thus, the aim of this study was to assess the association between UPF consumption and metabolic control in patients with T2DM.

## 2. Methods

### 2.1. Design and Ethical Aspects

This secondary cross-sectional analysis utilized baseline data from a multicenter study entitled ‘Nutritional Strategy for Glycemic Control in Patients With Type 2 Diabetes Mellitus: NUGLIC Study’ [18], conducted by the Hcor Research Institute in collaboration with the Brazilian Ministry of Health through the Development Support Program for the Unified Health System—PROADI-SUS.

The NUGLIC study [18] is registered on the ClinicalTrials.gov website under the registration identifier NCT03793855. This is an interventional study, with an actual start date of 6 May 2019, primary completion date of 28 September 2021, and study completion date of 30 December 2022. A total of 370 participants were enrolled [18]. This study received approval from the Research Ethics Committees from the Federal University of Health Sciences of Porto Alegre (UFCSPA) and Irmandade da Santa Casa de Misericordia de Porto Alegre (ISCMPA) and is registered under the number CAEE 97000618.3.2007.5335.

All participants voluntarily provided written informed consent to participate in this study, in accordance with the Declaration of Helsinki. This study involved minimal risk and followed both national and international guidelines for research involving human subjects. This study adhered to the STROBE Statement Checklist for Cross-Sectional Studies [19].

### 2.2. Participants

This study included individuals diagnosed with T2DM, aged 30 years or older, with glycated hemoglobin (HbA1C) levels of ≥7%, who had not received nutritional guidance for at least 6 months. Recruitment occurred among outpatients at six specialized endocrinology and nutrition referral services in the southern, southeastern, and northeastern regions of Brazil. Exclusion criteria comprised individuals with (a) type 1 diabetes mellitus (T1DM), latent adult autoimmune diabetes (LADA), or HbA1c ≥ 12%; (b) severe neuropathy; (c) chronic kidney disease; (d) a diagnosis of cancer or a life expectancy less than six months; (e) chemical dependency/alcoholism or the use of antipsychotics; (f) autoimmune disease or chronic steroid use; (g) gastroparesis; (h) pregnancy, lactation, or gestational DM; (i) an episode of acute coronary syndrome (ACS) within the past 60 days; (j) wheelchair use; (k) extreme obesity (body mass index [BMI] ≥ 40 kg/m^2^); (l) cognitive, neurological, or psychiatric conditions preventing participation in the study; and (m) involvement in other clinical intervention studies.

### 2.3. Measurements

The evaluation of patients followed a standardized protocol administered by appropriately trained and qualified researchers. Data were collected between the years 2019 and 2020. All patients underwent data collection related to anthropometric and biochemical indicators. Dietary consumption data, as well as sociodemographic and behavioral information such as gender, were also gathered, age (in years), race/ethnicity (white, black, yellow, mixed-race, or indigenous), marital status (single, married, divorced, in a stable union, or widowed), education level (illiterate and incomplete elementary school, incomplete elementary I and II, incomplete elementary school II and incomplete high school, complete high school and incomplete college education, or complete college education), pre-existing conditions (systemic arterial hypertension, dyslipidemia, acute myocardial infarction, and/or retinopathy), current medications, and smoking status (yes or no) were obtained through self-reports.

The assessment of anthropometric indicators was conducted by measuring weight (kg) and height (m) and waist circumference (WC) (cm), with the latter measured in duplicate, and the average value used for analysis. Weight and height were measured using appropriate attire, without accessories or shoes. These data were subsequently utilized for calculating Body Mass Index (BMI) (kg/m^2^). WC was measured at the midpoint between the lower edge of the ribcage and the iliac crest on the mid-axillary line [20,21].

The biochemical assessment included the analysis of total cholesterol (TC) (mg/dL), HDL-cholesterol (HDL-c) (mg/dL), LDL-cholesterol (LDL-c) (mg/dL), fasting triglycerides (TG) (mg/dL), fasting glucose (FG) (mg/dL), and HbA1c (%). This analysis was performed in reference laboratories of participating centers, using appropriate analytical techniques. Altered laboratory parameters were considered [22,23], with values: TC > 200 mg/dL, HDL-c < 40 mg/dL, LDL-c ≥ 100 mg/dL, TG ≥ 150 mg/dL, FG ≥ 130 mg/dL, and HbA1c ≥ 7.0%.

### 2.4. Exposure

The exposure variable was the consumption of processed foods, achieved by grouping the categories of PF and UPF. This grouping was necessary due to the relatively low consumption of ultra-processed foods observed in this sample. This consumption was measured as a percentage of total energy intake (%TEI) and categorized into quintiles.

Dietary intake was assessed through 24 h dietary recalls (R24h) at two different time points on random days: during the initial interview and seven days later through telephone calls. The estimation of macronutrients and micronutrients from each R24h was calculated using Vivanda^®^ software, allowing for the standardization of consumed food items.

The NOVA Food Classification [10,11] was utilized to categorize consumed foods into four processing groups: (1) fresh and minimally processed foods, (2) culinary ingredients, (3) PF, and (4) UPF. This classification system facilitates the grouping of foods based on the level and type of industrial processing they undergo, taking into account all physical, biological, and chemical methods employed, as well as the addition of synthetic additives for commercial purposes, and their impact on the food matrix and nutrient synergy [10,24].

Subsequently, the average energy intake for each NOVA group was calculated for each individual based on the two R24h collected. The resulting average energy intake for each group was then converted into a percentage of the TEI.

### 2.5. Outcome

Glycated hemoglobin (HbA1c) (%) was used as the primary outcome measure, while the metabolic parameters total cholesterol (TC) (mg/dL), high-density lipoprotein cholesterol (HDL-c) (mg/dL), low-density lipoprotein cholesterol (LDL-c) (mg/dL), triglycerides (TG) (mg/dL), fasting glucose (FG) (mg/dL), and glycated hemoglobin (HbA1c) (%), as well as Body Mass Index (BMI) (kg/m^2^) [20] and waist circumference (WC), were utilized as secondary outcome measures and treated as continuous variables. Obesity was defined as a categorical variable when the BMI was ≥30 kg/m^2^ and it was also treated as a secondary outcome.

### 2.6. Statistical Analysis

The sample size was calculated expecting a Cohen’s effect size of f = 0.20, suggested as a good initial estimate to obtain a non-negligible result of the difference in HB1AC between UPF quintiles with a significance of 5% and a power of 80% [25], resulting in 305 participants.

The analyses were conducted using the statistical software SPSS Version 25.0, considering results significant at *p* < 0.05. To assess the distribution of health, sociodemographic, anthropometric, dietary consumption, and behavioral characteristics of the sample, qualitative variables were presented as absolute and relative frequencies, while quantitative variables were reported as mean and standard deviation or the median and interquartile range (IQR). Normality was assessed using the Kolmogorov–Smirnov test.

The consumption of PF and UPF was combined into a single category and divided into quintiles. Differences between quintiles were assessed using chi-square tests with adjusted standardized residuals and the Kruskal–Wallis test with Bonferroni correction for multiple comparisons.

The impact of food consumption quintiles on categorical outcomes (obesity) was assessed using Poisson regression analysis, while multiple linear regression analysis was employed for all other outcomes, with adjustment for robust variance. All models were adjusted for age, race, sex, income, education, current medication, level of physical activity, and smoking status. Estimates of regression coefficients and odds ratios (OR) with 95% confidence intervals (CI) were presented.

## 3. Results

Sociodemographic, clinical, and therapeutic characteristics are outlined in Table 1. Out of the 370 participants initially included in the NUGLIC study [18], 44 were excluded from this analysis due to missing data on food consumption, exclusively resulting from their persistent unavailability for telephone contact during the COVID-19 pandemic. There were no significant differences in baseline characteristics between excluded and included participants; therefore, this exclusion is unlikely to have introduced relevant selection bias.

The final sample consisted of 326 participants, with 60% female, 63.1% married or in a stable relationship, 49.2% white, and 15% active smokers. The most prevalent comorbidities were hypertension (82.5%) and dyslipidemia (63.1%), while the most commonly used medications were biguanides (86.5%) and insulin (43.4%).

Table 2 presents the data from laboratory tests, anthropometric measurements, and dietary intake. The mean BMI was 30.3 ± 4.6 kg/m^2^, and WC was 103.1 ± 11.7 cm. Laboratory tests indicated a mean FG of 166.5 ± 59.3 mg/dL and HbA1c of 8.7 ± 1.5%. The mean daily energy intake was 1515.50 ± 598.74 kilocalories (kcal), with the highest dietary consumption represented by fresh or minimally processed foods (median of 64.4; IQR 54.1–73.5), while culinary ingredients reached a median of 1.8 (0–5.3). No extreme daily energy intake (<500 kcal/day or >6000 kcal/day) [26,27] was observed, which would necessitate excluding the data. The dietary consumption profile by processing group in each of the PF + UPF quintiles can be visualized in Table 3. From quintiles 1 to 5, while the consumption of fresh and minimally processed foods decreases, the consumption of PF and UPF increases, all following a linear trend. The consumption of culinary ingredients did not show the same trend. Even in the highest quintile of PF + UPF consumption, the median consumption of fresh and minimally processed foods (45.2; IQR 38.9–48.8) remained substantially higher than the median consumption of UPF (26.2; IQR 16.8–39.1).

In the multiple linear regression analysis, PF + UPF consumption in quintiles 3, 4, and 5 had a similar impact on TC and LDL-c levels across these quintiles, with some imprecision in the estimates. In quintiles 3 (Q3) and 4 (Q4) (Q3: 27.92–36.86% of TEI; Q4: 36.87–45.55% of TEI), the consumption of these foods was associated with the greatest observed increase in TC levels (Q3: 26.6 mg/dL; 95% CI 10.7–42.6; Q4: 26.7 mg/dL; 95% CI 10.69–42.69), compared to the consumption of individuals in the lowest quintile (Q1 ≤ 19.54% of TEI). Regarding the LDL-c fraction, the greatest impact of PF + UPF consumption was observed in Q4, being associated with an increase of 19.8 mg/dL (95% CI 6.93–32.67), compared to the reference category. However, the increases in TC and LDL-c levels did not follow a consistent linear trend across the quintiles of PF + UPF consumption (Table 4).

## 4. Discussion

In this study, we investigated the association of dietary intake, categorized according to the NOVA Food Classification, with the glycemic, lipid profile, and anthropometric data of patients with T2DM. Our findings revealed a predominant consumption of fresh or minimally processed foods, which accounted for 64% of the total daily energy intake. PF and UPF, as assessed by R24h, had a median consumption of 12.5% and 16.4% of TEI, respectively. We analyzed outcomes related to glycemic and lipid profiles, BMI, and WC across each quintile of PF and UPF consumption. However, only TC and LDL-c demonstrated an association with the highest quintiles of consumption (Q3, Q4, and Q5). For HDL-c, an isolated association was observed with the quintile of highest consumption (Q5).

The biological plausibility for foods to be associated with certain outcomes and not others in individuals with T2DM can be explained by a series of factors. These range from the nutritional composition of UPF, which, being rich in sugars, saturated and trans fats, sodium, and food additives, can directly affect the body’s metabolic response, influencing blood glucose levels and insulin sensitivity [28]. Additionally, their high glycemic index can lead to glucose spikes and subsequent insulin resistance in susceptible individuals, contributing to the development of type 2 diabetes and other adverse metabolic outcomes [29]. UPF can also promote chronic low-grade inflammation and oxidative stress in the body due to the presence of pro-inflammatory ingredients. Although these inflammatory and oxidative processes are implicated in the pathogenesis of type 2 diabetes mellitus (T2DM) and its related complications [30], we did not observe an association in our sample between the intake of processed and ultra-processed foods (PF + UPF) and elevated glycemic levels. It is possible that applying the R24 only twice, as well as the change in data collection method (from in-person to telephone interviews), may have affected the assessment of intake. Moreover, we cannot rule out that the COVID-19 pandemic may have led individuals to eat more meals at home, which could have influenced their dietary patterns [31] and possibly the glycemic levels observed in this sample.

The association between UPF and adverse outcomes in T2DM is multifaceted and involves a complex interaction between nutritional composition, glycemic index, inflammatory response, and oxidative stress. Individuals with T2DM, including those at increased risk of developing it, are particularly susceptible to the deleterious effects of these foods due to their specific pathophysiological characteristics [17]. Therefore, despite our findings, we consider that reducing the intake of UPF may be an important strategy in the prevention and management of T2DM and its associated complications.

In our sample, the median consumption of UPF was 16.4% of the TEI, which aligns with findings from other studies involving the Brazilian population in various settings. A cross-sectional analysis of baseline data from the ELSA-Brasil study (2008–2010) [32] reported mean UPF consumption values of 22.7% in a sample of adults. Similarly, a study by Canhada et al. [33] also conducted with participants from the ELSA study, reported a value of 24.6% for UPF consumption.

The relationship between UPF consumption and overweight and obesity has been described in studies involving different populations [32,33,34]. Indeed, UPF has a higher caloric content and when consumed excessively and/or within an unbalanced diet, they contribute to weight gain. Evidence suggests that high UPF consumption is associated with higher BMI and increased prevalence of overweight or obesity [35,36]. A cohort study involving 8451 Spanish adult university students reported that participants with the highest UPF consumption (Q4) had a higher risk of developing overweight or obesity (adjusted hazard ratio (HR) 1.26; 95% CI: 1.10–1.45; *p* = 0.001) compared to those in the lowest consumption quartile [34]. A similar finding was observed in a cross-sectional study [32], where the fourth quartile (>30.8%) versus the first quartile (<17.8%) of UPF consumption was associated with a 27% and 33% higher risk of weight gain and waist circumference, respectively. Similarly, those in the fourth quartile of consumption had a 20% higher risk of incident overweight/obesity and a 2% higher risk of incident obesity [32]. In our sample, there was no association between BMI or WC and the quintiles of higher PF + UPF consumption; however, unlike our data, the reported studies conducted dietary intake analysis through food frequency questionnaires.

The management of glycemic control in patients with T2DM is a cornerstone of non-pharmacological care as emphasized by guidelines [37]. Given that diet plays a protective role and is integral to the treatment for individuals with T2DM, appropriate nutrition focuses on a balanced consumption of whole foods, fiber, fruits, and vegetables. However, there is growing evidence indicating that the consumption of UPF is associated with various negative outcomes, including T2DM [13,38,39]. A systematic review with meta-analysis involving 415,554 participants (including 21,932 cases of T2DM) suggested that each 10% increase in UPF consumption was associated with a 12% higher risk (95% CI 10–13%) of developing T2DM [38].

The same study, utilizing data from three other large studies (the Nurses’ Health Study, Nurses’ Health Study II, and Health Professional Follow-Up Study), investigated the incidence of T2DM in relation to the consumption of UPF. Dietary intake was assessed using food frequency questionnaires, and UPF were categorized according to the NOVA Food Classification. The extreme quintiles of UPF intake demonstrated a HR of 1.56 (95% CI 1.47–1.65; *p* < 0.0001) for T2DM. Additionally, each increase of one serving per day was associated with a 5% higher risk (95% CI 5–6%) of developing T2DM [38].

Equally concerning is the impact of UPF consumption on the development of metabolic syndrome, a condition associated with an increased risk of T2DM [40]. Recent evidence from a study involving 8065 adults with a median UPF consumption of 360 g/day, followed for an average of 7.9 years (SD 1.3), revealed that 31.1% (2508) of individuals developed metabolic syndrome (MS). Through statistical modeling adjusted for TEI, socioeconomic factors, and behavioral factors such as smoking, alcohol consumption, and physical activity, it was observed that a 150 g/day increase in UPF consumption resulted in a 7% higher risk (95% CI 1.05–1.08) of developing MS.

Similarly, when consumption was analyzed by quartiles, those corresponding to the highest intakes, namely Q4 (between 366 g/day and 552 g/day) and Q4 (>552 g/day), demonstrated a risk of 19% (95% CI 1.08–1.32) and 33% (95% CI 1.20–1.47), respectively, for MS compared to the lowest consumption quartile [27]. These findings are consistent with those of studies conducted in other populations, which have shown that higher UPF intake is associated with a higher prevalence of MS, higher LDL-c levels, and elevated FG [41].

In relation to laboratory outcomes, we found that after adjusting for confounding factors, levels of TC and LDL-c were associated with the highest quintiles of PF + UPF consumption. When compared to the reference category, while the PF + UPF consumption in Q3 and Q4 was associated with a similar increase of 26.6 mg/dL and 26.7 mg/dL, respectively, in TC levels (Q3: 95% CI 10.7–42.6; Q4: 95% CI 10.69–42.69), consumption in Q4 was associated with an increase of 19.8 mg/dL (95% CI 6.93–32.67) in LDL-c levels.

Thus far, the scientific literature examining the association between UPF intake and lipid profile in individuals with T2DM remains limited. However, it has been previously suggested that a consumption of at least 700 g/day of UPF seems to impact a reduction of approximately 0.6 mg/dL in HDL-c (*p* = 0.13) and a consistent increase of 2 mg/dL in TG (*p* = 0.26) [27]. Positively, our sample of individuals with T2DM, who were not undergoing any nutritional monitoring, demonstrated a higher percentage of consumption of minimally processed or unprocessed foods, which may have influenced the observed lipid profile.

Possible limitations include the cross-sectional design itself, which does not account for the temporality between exposure and outcome and is therefore subject to reverse causality bias. Additionally, associations identified in cross-sectional studies do not imply causality. Issues related to the COVID-19 pandemic, which necessitated adapting data collection to a virtual mode, also represent potential limitations. However, this study focused on training researchers who conducted the collections and tabulation to minimize potential errors, along with verifying data quality to ensure reliable data reflecting reality. Due to the pandemic, R24h were conducted on only two days, with one of the days being via telephone, which warrants caution when interpreting the findings. Although R24h are among the most widely used methods in epidemiological studies, their retrospective nature is a known limitation that can lead to recall bias, a type of information bias [42]. Furthermore, this was a sub-study developed from the NUGLIC study [18], a multicenter randomized clinical trial originally designed to address a different research question. Nevertheless, it is worth noting that our study was conducted across multiple outpatient centers, providing a representative sample.

## 5. Conclusions

In our sample of patients with T2DM, higher consumption of PF + UPF was associated with higher levels of TC and LDL-c, which are important markers of cardiovascular risk. In contrast, no associations were observed between PF + UPF consumption and FG, HbA1c, BMI, or WC. These patients showed a lower daily energy intake from UPF compared to previous studies, which suggests a need for caution when interpreting these results but also highlights the importance of expanding strategies that promote the consumption of and access to fresh or minimally processed foods. Future longitudinal studies are needed to understand how the relationships between UPF consumption and metabolic parameters are established in individuals with T2DM, filling the gaps that still exist in the scientific literature. The cross-sectional nature of this study does not support any causal relationship.

## Figures and Tables

**Table 1 ijerph-22-01275-t001:** Sociodemographic, clinical, therapeutic, and laboratory characteristics of individuals with T2DM, Brazil 2021.

Variables	*n* = 326
Age (years)—mean ± SD	60.9 ± 9.3
Sex—*n* (%)	
Male	131 (40)
Female	195 (60)
Marital status—*n* (%)	
Single	54 (16.6)
Married/stable union	205 (63.1)
Divorced/widowed	66 (20.3)
Race/ethnicity—*n* (%) (*n* = 325)	
White	160 (49.2)
Black	74 (22.8)
Mixed-race	84 (25.8)
Yellow/indigenous	7 (2.2)
Education level—*n* (%)	
Illiterate and incomplete elementary school	90 (27.7)
Incomplete elementary I and II	69 (21.2)
Incomplete elementary school II and high school	55 (16.9)
Complete high school and incomplete college education	87 (26.8)
Complete college education	24 (7.4)
Most prevalent pre-existing conditions—*n* (%)	
Systemic arterial hypertension	268 (82.5)
Dyslipidemia	205 (63.1)
Acute myocardial infarction	64 (19.7)
Diabetic retinopathy	45 (13.8)
Current smokers—*n* (%)	15 (4.6)
Current medications—*n* (%)	
Sulfonylurea	119 (36.6)
Biguanides	281 (86.5)
SGLT2 inhibitors	32 (9.8)
Insulin	141 (43.4)
Others	87 (26.8)
Laboratory parameters—mean ± SD	
TC (mg/dL) (*n* = 323)	177.9 ± 46.4
LDL-c (mg/dL) (*n* = 318)	94.7 ± 37.8
HDL-c (mg/dL) (*n* = 321)	50.9 ± 16.4
TG (mg/dL) (*n* = 318)	164.8 ± 111.1
FG (mg/dL) (*n* = 321)	166.3 ± 59.3
HbA1c (%) (*n* = 320)	8.7 ± 1.5

SD: standard deviations; TC: total cholesterol; LDL-c: LDL-cholesterol; HDL-c: HDL-cholesterol; TG: fasting triglycerides; FG: fasting glucose; HbA1c: glycated hemoglobin.

**Table 2 ijerph-22-01275-t002:** Anthropometric data and food consumption of individuals with DM2, Brazil, 2021.

Variables	*n* = 326
	Mean ± SD/Median (IQR)
Anthropometric data	
Weight (kg)	79.8 ± 14.1
BMI (kg/m^2^)	30.3 ± 4.6
WC (cm)	103.1 ± 11.7
Male (*n* = 129)	105.8 ± 11.6
Female (*n* = 194)	103.7 ± 34.9
Food consumption (*n* = 325)	
Daily energy intake (kcal)	1515.50 ± 598.74
NOVA Food Classification (*n* = 325)	
Fresh and minimally processed foods (% kcal)	64.4 (54.1–73.5)
Culinary ingredients (% kcal)	1.8 (0–5.3)
Processed foods (% kcal)	12.5 (5.8–22)
Ultra-processed foods (% kcal)	16.4 (8.9–25.5)

SD: standard deviations; IQR: interquartile ranges; BMI: Body Mass Index; WC: waist circumference; kg: kilograms; kcal: kilocalories.

**Table 3 ijerph-22-01275-t003:** Food consumption according to processed and ultra-processed foods quintiles in individuals with DM2, Brazil, 2021.

	Quintiles of Processed and Ultra-Processed Food Consumption (% of Total Energy)					
**Food Group**	Q1 ≤19.54%	Q2 19.55–27.91%	Q3 27.92–36.86%	Q4 36.87–45.55%	Q5 >45.55%	***p*-Value**
	**Median (IQR)**	**Median (IQR)**	**Median (IQR)**	**Median (IQR)**	**Median (IQR)**	
Fresh and minimally processed foods	83.8 (78; 87.4)	72.2 (69.9; 74.1)	65.0 (62.7; 68)	57.5 (54.8; 59.5)	45.2 (38.9; 48.8)	<0.001
Culinary ingredients	2.4 (0; 6.1)	2.8 (0; 6)	2.0 (0.1; 4.3)	0.6 (0; 4.4)	1.3 (0; 3.4)	<0.001
Processed foods	5.7 (0; 8.1)	11.0 (6.3; 17.6)	13.2 (5.4; 21.9)	17.8 (12; 27.8)	26.0 (12.7; 39.9)	0.039
Ultra-processed foods	8.2 (3.3; 12.1)	13.4 (8; 17.7)	18.5 (10.9; 27.3)	23.5 (12.3; 29.1)	26.2 (16.8; 39.1)	<0.001

Q1, first quintile; Q2, second quintile; Q3, third quintile; Q4, fourth quintile; Q5, fifth quintile. *p*-values according to the Kruskal–Wallis test (continuous variables).

**Table 4 ijerph-22-01275-t004:** Multivariable analysis of the association between the quintile-relative dietary contribution of processed and ultra-processed foods (% of total energy) and metabolic parameters in individuals with DM2, Brazil, 2021.

Metabolic and Anthropometric Parameters	Q1 ≤19.54%		Q2 19.55–27.91%		Q3 27.92–36.86%		Q4 36.87–45.55%		Q5 >45.55%	
	β/OR	95% CI	β/OR	95% CI	β/OR	95% CI	β/OR	95% CI	β/OR	95% CI
FG (mg/dL) (β)	0-0	Ref.	7.93	−13.35; 28.1	14.52	−5.99; 35.03	12.59	−7.99; 33.17	15.60	−5.14; 36.34
HbA1c (%) (β)	0-0	Ref.	−0.17	−0.66; 0.33	−0.01	−0.55; 0.48	0.09	−0.41; 0.58	0.16	−0.34; 0.66
TC (mg/dL) (β)	0-0	Ref.	9.71	−6.37; 25.79	26.6 **	10.7; 42.6	26.7 **	10.69; 42.69	22.5 *	6.36; 38.64
LDL-c (mg/dL) (β)	0-0	Ref.	7.25	−5.7; 20.19	17.5 **	4.76; 30.34	19.8 **	6.93; 32.67	17.5 *	4.51; 30.45
HDL-c (mg/dL) (β)	0-0	Ref.	4.28	−0.69; 9.25	2.59	−2.34; 7.51	4.17	−0.78; 9.11	5.07	0.08; 10.06
TG (mg/dL) (β)	0-0	Ref.	−1.92	−41.11; 37.26	29.5	−9.27; 68.27	10.69	−28.18; 49.56	2.61	−36.44; 41.65
BMI (kg/m^2^) (β)	0-0	Ref.	0.02	−1.57; 1.6	−0.25	−1.82; 1.33	−0.48	−2.06; 1.1	−0.92	−2.51; 0.66
WC (cm) (β)	0-0	Ref.	1.38	−2.66; 5.42	−0.43	−4.44; 3.57	−1.5	−5.51; 2.52	−1.34	−5.38; 2.69
Obesity (BMI ≥ 30 kg/m^2^) (OR)	1-0	Ref.	1.00	0.63; 1.59	0.84	0.52; 1.34	0.85	0.53; 1.37	0.91	0.57; 1.46

* *p*-value < 0.05; ** *p*-value < 0.01. Q1, first quintile; Q2, second quintile; Q3, third quintile; Q4, fourth quintile; Q5, fifth quintile. Ref.: reference group; FG: fasting glucose; HbA1c: glycated hemoglobin; TC: total cholesterol; LDL-c: LDL-cholesterol; HDL-c: HDL-cholesterol; TG: fasting triglycerides; BMI: Body Mass Index; WC: waist circumference; kg: kilograms. Adjusted for age, race, sex, income, education, current medication, level of physical activity, and smoking status.

## Data Availability

The data used in this study are available from the corresponding author upon reasonable request.

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
