# Peer review of "Consumption of Ultra-Processed Foods and Metabolic Parameters in Type 2 Diabetes Mellitus: A Cross-Sectional Study"

_ijerph, 2025, doi:10.3390/ijerph22081275_

Round 1

Reviewer 1 Report

Comments and Suggestions for Authors

The manuscript presents a cross-sectional analysis from the baseline data of the NUGLIC study, evaluating the association between ultra-processed food (UPF) consumption and metabolic parameters in individuals with type 2 diabetes mellitus (T2DM). The study is relevant and timely, considering the growing interest in the impact of food processing on chronic diseases. The use of the NOVA classification and a sizable Brazilian cohort provides a solid foundation for the investigation. However, the study has some limitations related to statistical interpretation, clarity in writing, and scientific rigor in hypothesis development. The manuscript would benefit from major revisions before being considered for publication. Scientific Quality Strengths: Use of a multicenter cohort and real-world data. Application of the NOVA food classification, an increasingly relevant tool in nutritional epidemiology. Adjustment for key confounders in multivariable analyses. Ethical and methodological transparency. Weaknesses: The cross-sectional design limits causal inference, which should be more clearly acknowledged in the discussion and abstract. Some results (e.g., lack of association with HbA1c and BMI) are inconclusive and may reflect measurement limitations. The dietary intake was assessed using only two 24-hour recalls, which may not be adequate to capture usual intake, especially for UPFs. Combining processed and ultra-processed foods into a single exposure variable might dilute specific associations and misrepresent the role of UPFs alone. 3. Specific Comments Title and Abstract The title is appropriate and informative. The abstract accurately reflects the study design and results but should explicitly mention the cross-sectional nature to temper interpretation. Introduction Well-structured and provides an adequate rationale. Consider refining the hypothesis to differentiate more clearly between processed and ultra-processed foods. Methods The description is mostly thorough. There should be more justification for combining PF and UPF into a single exposure variable. Consider analyzing them separately to provide a more granular interpretation. Detail how missing dietary data led to the exclusion of 44 participants—was there any bias in this exclusion? Results The data is well presented, but the interpretation in the text should match the tables more closely. Consider adding interaction analyses (e.g., gender, medication use) if possible. Discussion The discussion is rich in contextual information but tends to overstate the results. For example, there is no statistically significant association with HbA1c, yet the text implies a broader metabolic implication. Better contextualize your findings within existing literature, particularly studies that observed stronger links between UPF and glycemic outcomes. Discuss potential measurement errors in dietary assessment and their implications more fully. Conclusions The conclusion is consistent with the findings but should better reflect the limited associations observed and avoid causal language. 4. Recommendations for Improvement Separate analysis of processed and ultra-processed food impacts if feasible. Clarify limitations related to dietary assessment, residual confounding, and cross-sectional design. Rephrase some conclusions to avoid suggesting causality. Expand on public health implications only to the extent justified by the data. Improve clarity and grammar throughout the manuscript—some sentences are overly long or ambiguous.

Author Response

Weaknesses

The cross-sectional design limits causal inference, which should be more clearly acknowledged in the discussion and abstract. 

Response: We appreciate your suggestion and have highlighted the limitation of not being able to establish causal inference with a cross-sectional study design. The following correction can be found in lines 40–41: ‘Due to the cross-sectional design of this study, these findings do not support a causal relationship’.

Some results (e.g., lack of association with HbA1c and BMI) are inconclusive and may reflect measurement limitations. The dietary intake was assessed using only two 24-hour recalls, which may not be adequate to capture usual intake, especially for UPFs.

Response: Thank you for your thoughtful suggestion. We clarified the limitations related to the 24-hour dietary recall (R24h) in the limitations paragraph of the ‘Discussion section’. The revised text can be found in lines 361–364: ‘Due to the pandemic, R24h were conducted on only two days, with one of the days conducted via telephone, which warrants caution when interpreting the findings. Although R24h are among the most widely used methods in epidemiological studies, their retrospective nature is a known limitation that can lead to recall bias, a type of information bias (42)’.

Combining processed and ultra-processed foods into a single exposure variable might dilute specific associations and misrepresent the role of UPFs alone.

Response: Thank you very much for your pertinent observation. We agree that conducting separate analyses for ultra-processed and processed foods could contribute to the proposed investigation. However, the proportion of ultra-processed food consumption in this sample was too low to support stratified analysis. Therefore, in the researchers’ view, combining ultra-processed and processed food groups represented a more informative analytical approach in this study.

Specific Comments

The abstract accurately reflects the study design and results but should explicitly mention the cross-sectional nature to temper interpretation.

Response: We appreciate your suggestion and have highlighted the limitation of not being able to establish causal inference with a cross-sectional study design. The following correction can be found in lines 40–41: ‘Due to the cross-sectional design of this study, these findings do not support a causal relationship’.

Introduction

Consider refining the hypothesis to differentiate more clearly between processed and ultra-processed foods.

Response: Thank you for your valuable comment. Please note that the original hypothesis refers specifically to the consumption of ultra-processed foods, in line with our research objective. However, given the consumption profile of this sample, it was necessary to combine ultra-processed and processed foods to address the research question. To address this issue, we chose to provide a transparent description of the analytical decision and to highlight it as a study limitation.

Methods

The description is mostly thorough. There should be more justification for combining PF and UPF into a single exposure variable. Consider analyzing them separately to provide a more granular interpretation.

Response: Thank you for your valuable comment. We have included a justification for grouping the processed and ultra-processed food categories. The revised sentence can be found in lines 148–150: ‘The exposure variable was the consumption of processed foods, created by grouping the categories of processed foods and ultra-processed foods. This grouping was necessary due to the relatively low consumption of ultra-processed foods observed in this sample. Consumption was measured as a percentage of total energy intake (%TEI) and categorized into quintiles’. 

As mentioned above, the proportion of ultra-processed food consumption in this sample was too low to support a stratified analysis. Therefore, in the researchers’ view, combining ultra-processed and processed food groups provided a more informative analytical approach in this study.

Detail how missing dietary data led to the exclusion of 44 participants—was there any bias in this exclusion? 

Response: Thank you for your comment. We excluded 44 participants due to missing dietary data, as these participants were persistently unavailable for telephone contact during the COVID-19 pandemic. We compared the excluded and included participants and found no significant differences in baseline variables. Therefore, we believe this exclusion is unlikely to have introduced relevant selection bias. These details have been included in the manuscript, lines 199–204: ‘Out of the 370 participants initially included in the NUGLIC study (18), 44 were excluded from this analysis due to missing data on food consumption, exclusively resulting from their persistent unavailability for telephone contact during the COVID-19 pandemic. There were no significant differences in baseline characteristics between excluded and included participants; therefore, this exclusion is unlikely to have introduced relevant selection bias’.

Results 

The data is well presented, but the interpretation in the text should match the tables more closely. 

Response: Thank you for the suggestion. We sought to bring the description of the results closer to what is presented in the table, including highlighting some of the limitations and uncertainties of the estimates that were not previously discussed. Please find below the corrections: 

Lines 234-236: ‘In the multiple linear regression analysis, PF + UPF consumption in quintiles 3, 4, and 5 had a similar impact on TC and LDL-c levels across these quintiles, with some imprecision in the estimates’.

Lines 242-243: ‘However, the increases in TC and LDL-c levels did not follow a consistent linear trend across the quintiles of PF + UPF consumption’.

Consider adding interaction analyses (e.g., gender, medication use) if possible. 

Response: Thank you for your valuable comment. We agree that planning an a priori interaction analysis could have been relevant, provided the necessary assumptions were considered. However, we understand that introducing it at this stage, without sufficient statistical power to support it, would not yield reliable results. As recognized in the scientific literature, including interaction terms in a regression model requires greater statistical power than tests of main effects, and in small samples these analyses are often uninformative. Given that our estimates showed some imprecision, as evidenced by the wide confidence intervals—which are closely related to sample size—we believe this is a clear indication that our sample and statistical power are insufficient to perform the suggested analysis.

VanderWeele TJ, Knol MJ. A Tutorial on Interaction. Epidemiologic Methods, vol. 3, no. 1, 2014, pp. 33-72. https://doi.org/10.1515/em-2013-0005

Jaccard J, Turrisi R. Interaction Effects in Multiple Regression. 2nd ed. Sage Publications; 2003.

Rimpler A, Kiers HAL, van Ravenzwaaij D. To interact or not to interact: The pros and cons of including interactions in linear regression models. Behav Res Methods. 2025;57(3):92. Published 2025 Feb 7. doi:10.3758/s13428-025-02613-6

Discussion 

The discussion is rich in contextual information but tends to overstate the results. For example, there is no statistically significant association with HbA1c, yet the text implies a broader metabolic implication. Better contextualize your findings within existing literature, particularly studies that observed stronger links between UPF and glycemic outcomes. Discuss potential measurement errors in dietary assessment and their implications more fully. 

Response: Thank you for your valuable contribution. We have made corrections to the text to incorporate your suggestion.

Lines 271-279: ‘Although these inflammatory and oxidative processes are implicated in the pathogenesis of type 2 diabetes mellitus (T2DM) and its related complications (30), we did not observe an association in our sample between the intake of processed and ultra-processed foods (PF + UPF) and elevated glycemic levels. It is possible that applying the R24 only twice, as well as the change in data collection method (from in-person to telephone interviews), may have affected the assessment of intake. Moreover, we cannot rule out that the COVID-19 pandemic may have led individuals to eat more meals at home, which could have influenced their dietary patterns (31) and possibly the glycemic levels observed in this sample’. 

Lines 284-286: ‘Therefore, despite our findings, we consider that reducing the intake of UPF may be an important strategy in the prevention and management of T2DM and its associated com-plications’

The limitations related to the dietary assessment method, as well as the specific challenges encountered in this study, were detailed in the paragraph on limitations in the discussion section, as reflected in the revisions indicated in the lines 357-364: ‘Issues related to the COVID-19 pandemic, which necessitated adapting data collection to a virtual mode, also represent potential limitations. However, the study focused on training researchers who conducted the collections and tabulation to minimize potential errors, along with verifying data quality to ensure reliable data reflecting reality. Due to the pandemic, R24h were conducted on only two days, with one of the days being via telephone, which warrants caution when interpreting the findings. Although R24h are among the most widely used methods in epidemiological studies, their retrospective na-ture is a known limitation that can lead to recall bias, a type of information bias (42)’.

Conclusions 

The conclusion is consistent with the findings but should better reflect the limited associations observed and avoid causal language. 

Response: Thank you for your pertinent observation. We have revised the conclusion to incorporate this suggestion while maintaining clarity regarding the limitations of our findings. In addition, the conclusion section has been highlighted as a subsection. The changes can be found in lines 372–382, as described below: ‘In our sample of patients with T2DM, higher consumption of PF + UPF was associated with higher levels of TC and LDL-c, which are important markers of cardiovascular risk. In contrast, no associations were observed between PF + UPF consumption and FG, HbA1c, BMI, or WC. These patients showed a lower daily energy intake from UPF com-pared to previous studies, which suggests caution when interpreting these results but also highlights the importance of expanding strategies that promote the consumption of and access to fresh or minimally processed foods. Future longitudinal studies are needed to understand how the relationships between UPF consumption and metabolic parameters are established in individuals with T2DM, filling the gaps that still exist in the scientific literature. The cross-sectional nature of this study does not support any causal relationship’.

Recommendations for Improvement 

Separate analysis of processed and ultra-processed food impacts if feasible. 

Thank you very much for your pertinent observation. We agree that conducting separate analyses for ultra-processed and processed foods could contribute to the proposed investigation. However, the proportion of ultra-processed food consumption in this sample was too low to support stratified analysis. Therefore, in the researchers’ view, combining ultra-processed and processed food groups represented a more informative analytical approach in this study.

Clarify limitations related to dietary assessment, residual confounding, and cross-sectional design. 

Response: Thank you very much for your valuable feedback. We made revisions to the limitations paragraph to incorporate these important points. The revised text can be found in lines 354–368: ‘Possible limitations include the cross-sectional design itself, which does not account for the temporality between exposure and outcome and is therefore subject to reverse cau-sality bias. Additionally, associations identified in cross-sectional studies do not imply causality. Issues related to the COVID-19 pandemic, which necessitated adapting data collection to a virtual mode, also represent potential limitations. However, the study focused on training researchers who conducted the collections and tabulation to minimize potential errors, along with verifying data quality to ensure reliable data reflecting reality. Due to the pandemic, R24h were conducted on only two days, with one of the days being via telephone, which warrants caution when interpreting the findings. Although R24h are among the most widely used methods in epidemiological studies, their retrospective na-ture is a known limitation that can lead to recall bias, a type of information bias (42). Furthermore, this was a sub-study developed from the NUGLIC study, a multicenter random-ized clinical trial originally designed to address a different research question. Nevertheless, it is worth noting that our study was conducted across multiple outpatient centers, providing a representative sample’.

Rephrase some conclusions to avoid suggesting causality. 

Response: We appreciate your suggestion and have emphasized, both in the abstract and in the conclusions section, the inability to establish causal inference due to the cross-sectional study design.

Expand on public health implications only to the extent justified by the data. 

Response: Thank you for your pertinent contribution. We revised the conclusion to include more objective considerations within the scope of public health. This change can be found in lines 375–378: ‘These patients showed a lower daily energy intake from UPF compared to previous stud-ies, which suggests caution when interpreting these results but also highlights the im-portance of expanding strategies that promote the consumption of and access to fresh or minimally processed foods’. 

Improve clarity and grammar throughout the manuscript—some sentences are overly long or ambiguous.

Thank you for your careful evaluation. We have thoroughly revised the manuscript, including improvements in grammar, cohesion, redundancy, and sentence ambiguity.

Reviewer 2 Report

Comments and Suggestions for Authors

The study focuses on observing the effect of processed and ultra-processed foods on metabolic parameters in people with Type 2 diabetes. The study utilizes a cross-sectional design to identify cause effect relationship between specific dietary intake and metabolic responses. Please consider the following suggestions to improve the impact of the article.

  1. Line 23 (Abstract): Study aim is very vague. Keep a standard aim consistent throughout the study.
  2. Line 26: Expand NUGLIC here as the abbreviation was first used here.
  3. Lin3 29: Detail the primary and secondary outcomes measure in the methods.
  4. Line 37 (conclusion): Statement is very generic. Identify the population of T2DM.
  5. Line 75: Ideally this paper provides evidence of UPF in disease etiology and progression. Incorporate this concept when speaking about lack of evidence in treating T2DM.
  6. Line 84: If a paper was published form the original study – NUGLIC study; please reference it. And under study design incorporate that study uses secondary data analysis.
  7. Line 208-209: Elaborate Table 3 beyond these 2 lines. Give brief highlights on quintile classification/definition. And describe briefly what table 3 is depicting.
  8. Table 3 & 4 are unclear. Formatting and table design need to be revisited. Table 4 – Without showing the statistical significance it’s hard to understand if the study’s predictor variables are significantly related to the outcomes measured.
  9. Line 340: If this is the conclusion section, please use the subheading. Also start with addressing the study aim followed by the rest of the discussion.

Author Response

Line 23 (Abstract): Study aim is very vague. Keep a standard aim consistent throughout the study.

Response: Thank you for your pertinent observation. We have revised the text to ensure consistency with the description provided in the manuscript. The following correction can be found in lines 22–24: ‘This study aimed to assess the association between ultra-processed food consumption, as classified by the NOVA Food Classification, and metabolic control in patients with type 2 diabetes mellitus’.

Line 26: Expand NUGLIC here as the abbreviation was first used here.

Response: Thank you for the suggestion. We have made the necessary correction, which can be found in lines 25–26 of the manuscript.

Line 29: Detail the primary and secondary outcomes measure in the methods.

Response: We appreciate the suggestion and have included a description of the outcomes measured. The following correction can be found in lines 27-30: ‘Multiple linear regression and Poisson regression were employed to evaluate the associa-tion between quintiles of processed and ultra-processed food consumption and glycated hemoglobin as the primary outcome, and fasting glucose, lipid profile, body mass index, and waist circumference as secondary outcomes’.

Line 37 (conclusion): Statement is very generic. Identify the population of T2DM.

Response: We appreciate your suggestion and have revised the text accordingly. Additionally, in this section, we have acknowledged the limitation of not being able to establish causal inference, as recommended by Reviewer 1. The following correction can be found in lines 37-41: ‘In patients with type 2 diabetes mellitus, higher consumption of processed and ultra-processed foods was associated with higher levels of total cholesterol and LDL cholesterol, which are important cardiovascular risk parameters. Due to the cross-sectional design of this study, these findings do not support a causal relationship’.

Line 75: Ideally this paper provides evidence of UPF in disease etiology and progression. Incorporate this concept when speaking about lack of evidence in treating T2DM.

Response: Thank you for your suggestion. We have revised the text and included a note highlighting the relevance of UPF consumption to T2DM progression. The following sentence can be found in lines 79–82: ‘This knowledge is also crucial for preventing disease progression and reducing mortality in T2DM. It is already known that, in this patient group, higher consumption of ultra-processed foods, compared to lower consumption, has been associated with a 64% and 155% higher risk of all-cause and cardiovascular mortality, respectively (17)’.

Line 84: If a paper was published form the original study – NUGLIC study; please reference it. And under study design incorporate that study uses secondary data analysis.

Response: Thank you for your valuable comment. We have implemented the suggested corrections, which can be found in lines 90 and 92. Additionally, the NUGLIC study has been properly referenced throughout the manuscript whenever cited.

Line 208-209: Elaborate Table 3 beyond these 2 lines. Give brief highlights on quintile classification/definition. And describe briefly what table 3 is depicting.

Response: Thank you for your suggestion. We have added some information while taking care not to repeat content already presented in the table, as we believe this could make the manuscript overly long and redundant.This revision can be found in lines 221–226: ‘From quintiles 1 to 5, while the consumption of fresh and minimally processed foods decreases, the consumption of PF and UPF increases, all following a linear trend. The consumption of culinary ingredients did not show the same trend. Even in the highest quintile of PF + UPF consumption, the median consumption of fresh and minimally processed foods (45.2; IQR 38.9–48.8) remained substantially higher than the median consumption of UPF (26.2; IQR 16.8–39.1)’.

Table 3 & 4 are unclear. Formatting and table design need to be revisited. Table 4 – Without showing the statistical significance it’s hard to understand if the study’s predictor variables are significantly related to the outcomes measured.

Response: Thank you for your valuable comment. We made adjustments to Tables 3 and 4 to improve clarity. For the construction of these tables, we used the format typically adopted in studies on food consumption stratified by percentiles. Some examples are cited below. Therefore, we believe that readers will be familiar with this table format. 

  • Juul F, Martinez-Steele E, Parekh N, Monteiro CA, Chang VW. Ultra-processed food consumption and excess weight among US adults. British Journal of Nutrition. 2018;120(1):90-100. doi:10.1017/S0007114518001046

  • Rauber F, Steele EM, Louzada MLDC, Millett C, Monteiro CA, Levy RB. Ultra-processed food consumption and indicators of obesity in the United Kingdom population (2008-2016). PLoS One. 2020;15(5):e0232676. doi:10.1371/journal.pone.0232676

Considering the large amount of data presented in Table 4, we chose to display the confidence intervals instead of p-values, in order to allow readers to also assess the precision of the estimates. The p-values were added at the end of the table, along with the corresponding legends.

Line 340: If this is the conclusion section, please use the subheading. Also start with addressing the study aim followed by the rest of the discussion.

Response: Thank you very much for your comment. We have incorporated the suggested changes. The following corrections can be found in lines 370–382.

‘5. Conclusion  

‘In our sample of patients with T2DM, higher consumption of PF + UPF was associated with higher levels of TC and LDL-c, which are important markers of cardiovascular risk. In contrast, no associations were observed between PF + UPF consumption and FG, HbA1c, BMI, or WC. These patients showed a lower daily energy intake from UPF compared to previous studies, which suggests caution when interpreting these results but also highlights the importance of expanding strategies that promote the consumption of and access to fresh or minimally processed foods’.
